# Development of Atomoxetine-Loaded NLC In Situ Gel for Nose-to-Brain Delivery: Optimization, In Vitro, and Preclinical Evaluation

**DOI:** 10.3390/pharmaceutics15071985

**Published:** 2023-07-20

**Authors:** Dibyalochan Mohanty, Omar Awad Alsaidan, Ameeduzzafar Zafar, Trishala Dodle, Jeetendra Kumar Gupta, Mohd Yasir, Anshuman Mohanty, Mohammad Khalid

**Affiliations:** 1Department of Pharmaceutics (Centre for Nanomedicine), School of Pharmacy, Anurag University, Hyderabad 500088, Telangana, India; trishasailoo61@gmail.com; 2Department of Pharmaceutics, College of Pharmacy, Jouf University, Sakaka 72341, Al-Jouf, Saudi Arabia; osaidan@ju.edu.sa (O.A.A.); azafar@ju.edu.sa (A.Z.); 3Institute of Pharmaceutical Research, GLA University Mathura, Chaumuhan 281406, Uttar Pradesh, India; jk.gupta@gla.ac.in; 4Department of Pharmacy, College of Health Sciences, Arsi University, Asella P.O. Box 396, Ethiopia; mohdyasir31@gmail.com; 5Product Development, Innovation and Science, Amway Global Services India Pvt. Ltd., Gurugram 122001, Haryana, India; anshumanmohanty@hotmail.com; 6Department of Pharmacognosy, College of Pharmacy, Prince Sattam Bin Abdulaziz University, Al-Kharj 11942, Saudi Arabia; drkhalid8811@gmail.com

**Keywords:** atomoxetine, dementia, nanostructured lipid carriers in situ gel, nose-to-brain deliver, pharmacokinetic and neuro-pharmacokinetic

## Abstract

The present study investigates the brain-targeted efficiency of atomoxetine (AXT)-loaded nanostructured lipid carrier (NLC)-laden thermosensitive in situ gel after intranasal administration. AXT-NLC was prepared by the melt emulsification ultrasonication method and optimized using the Box–Behnken design (BBD). The optimized formulation (AXT-NLC) exhibited particle size PDI, zeta potential, and entrapment efficiency (EE) of 108 nm, 0.271, −42.3 mV, and 84.12%, respectively. The morphology of AXT-NLC was found to be spherical, as confirmed by SEM analysis. DSC results displayed that the AXT was encapsulated within the NLC matrix. Further, optimized NLC (AXT-NLC13) was incorporated into a thermosensitive in situ gel using poloxamer 407 and carbopol gelling agent and evaluated for different parameters. The optimized in situ gel (AXT-NLC13G4) formulation showed excellent viscosity (2532 ± 18 Cps) at 37 °C and formed the gel at 28–34 °C. AXT-NLC13-G4 showed a sustained release of AXT (92.89 ± 3.98% in 12 h) compared to pure AXT (95.47 ± 2.76% in 4 h). The permeation flux through goat nasal mucosa of AXT from pure AXT and AXT-NLC13-G4 was 504.37 µg/cm^2^·h and 232.41 µg/cm^2^·h, respectively. AXT-NLC13-G4 intranasally displayed significantly higher absolute bioavailability of AXT (1.59-fold higher) than intravenous administration. AXT-NLC13-G4 intranasally showed 51.91% higher BTP than pure AXT (28.64%) when administered via the same route (intranasally). AXT-NLC13-G4 showed significantly higher BTE (207.92%) than pure AXT (140.14%) when administered intranasally, confirming that a high amount of the AXT reached the brain. With the disrupted performance induced by L-methionine, the AXT-NLC13-G4 showed significantly (*p* < 0.05) better activity than pure AXT as well as donepezil (standard). The finding concluded that NLC in situ gel is a novel carrier of AXT for improvement of brain delivery by the intranasal route and requires further investigation for more justification.

## 1. Introduction

Dementia is a progressive neurodegenerative syndrome. About 50 million people worldwide suffer from dementia. According to prevalence estimates, 12.5% of people over 60 years and 17.3% of people over 85 years of age have severe dementia, and both of these rates are projected to climb significantly over the next 25 years [1]. Dementia is characterized by loss of memory, cognitive and executive function, language, attention, and behavior. It interferes with the everyday activities of humans. In the case of vascular dementia, there is a compromised vascular system, which interrupts the adequate flow of blood and, consequently, disturbs the oxygen and nutrients supplied to the brain. All these issues may ultimately result in neuronal cell death [2,3]. Researchers established behavioral as well as biochemical traits linked to dementia of vascular origin using animal models with metabolic abnormalities [3]. Research has shown a strong link between cardiovascular disease, involving the heart and blood vessels, and cerebrovascular disease, involving the brain, and subsequent cognitive impairment and dementia [2,4]. Atomoxetine (AXT) is a secondary amino compound used to treat a wide range of neurological disorders and developmental difficulties. It is a selective norepinephrine reuptake inhibitor (NE-RI). Studies in rodents have shown that AXT enhances cognition, including memory, spatial recall, and executive function [5]. Endothelial health is linked to norepinephrine (NE). NE-RI lessens the effects of endothelial dysfunction and dementia. AXT has anti-oxidative actions, including protection from oxidative stress, apoptosis, inflammation, and cholinergic dysfunction. Even though ATX is an approved drug for the treatment of attention deficit hyperactivity disorder (ADHD) and vascular dementia, researchers have not focused on the application of AXT in such cases [6]. We postulated that AXT would have therapeutic effects on endothelial dysfunction and vascular-related dementia. AXT belongs to the BCS class I drug class, exhibiting high water solubility and high intestinal permeability [7]. It displayed bioavailability of 63% and 94% in its extensive and poor metabolizers, respectively [7]. To date, few formulations of AXT have been reported for nose-to-brain delivery, i.e., AXT-loaded solid lipid nanoparticles [7], AXT-loaded liposomes [8], and mucoadhesive thermosensitive gel [9]. Nose-to-brain delivery is a highly vascular, easy, and non-invasive route for drug delivery that can overcome first-pass metabolism. The nasal delivery has recently attracted a lot of attention [10] because of the special anatomical features of the nasal cavity, which allow the absorption of drugs through olfactory trigeminal areas and bypass the blood–brain barrier (BBB) [11]. There have been various studies that have revealed that the nasal route is an efficient way to administer the drug to the brain [12,13].

Lipid nanocarriers have been used to improve the bioavailability of certain drugs [14]. Additionally, lipid nanocarriers exhibited great potential for brain targeting due to their lipid content and nanosize, as they can passively diffuse directly across the BBB via the nasal route [15]. Various nanocarrier formulations have been reported for the delivery of drugs to the brain via the nasal route, i.e., liposome, polymeric nanocarrier, and solid lipid nanoparticle, but they have some limitations. Liposomes have major drawbacks, such as low drug loading and poor stability [16]. An alternative to both liposomes and polymeric nanocarriers, solid lipid nanoparticles (SLNs) are good carriers for drug delivery. It is more stable than liposomes and polymeric nanocarriers, has no or minimal requirement for organic solvent for preparation, and has a prolonged duration of action [17]. On the other hand, SLNs exhibited main drawbacks such as crystallization of solid lipids, which may lead to drug ejection and unequal drug distribution. In light of these shortcomings, nanostructured lipid carriers (NLCs) as second-generation SLNs have been proposed [18]. NLCs are a heterogeneous mixture of solid and liquid lipids. The lipids used for the development of NLCs are of fatty acid origin. It provides a large distance between the fatty acid chains of glycerides, leads to more drug loading in NLCs, and prevents drug expulsion during storage. NLCs have gained popularity as potential drug carriers for brain targeting, as their utility has already been explored for the delivery of certain classes of brain-acting drugs such as anticonvulsants, antipsychotics, antidepressants, anti-anxiety drugs, etc. [19]. The limitation of this route is that the chance of formulation elimination from the nasal cavity may be greater due to rapid mucous clearance as well as ciliary movement [20,21]. Both are responsible for the reduction in retention time in the nasal cavity. The small volume of each nostril, metabolic enzymes, and anatomical location of the olfactory epithelium are other main limitations of this route. So, the dosage form must first be able to reach this site [21]. Therefore, the drug delivery system needs to have retention capabilities to prevent rapid elimination. The mucoadhesive formulation can allocate the drug to the brain effectively and remains at the administration site for a long time [22]. In situ gel can be chosen as the best alternative among the conventional mucoadhesive dosage forms because it is present in the form of liquid (sol) and transforms into gel after administration to the nose by ionic interactions, changes in pH, temperature, etc. The higher viscosity of in situ gel not only reduces anterior and post-nasal leakage but also lessens irritation of the nasal mucosa by using excipients [22,23]. To date, a few studies have been reported for drug-loaded NLC in situ gels for brain targeting via the intranasal route [23,24,25,26]. The present study aimed to develop and characterize an AXT-loaded NLC in situ gel to improve AXT transport to the brain through intranasal delivery. AXT-NLCs were prepared by melt emulsification ultrasonication and optimized by Box–Behnken design (BBD). The optimized AXT-NLC was evaluated for particle size, PDI, zeta potential, entrapment efficiency, and morphology. Further optimized AXT-NLCs were incorporated into the thermosensitive gel and examined for pH analysis, viscosity, drug content, gelling temperature, in vitro drug release, and an ex vivo permeation study. Further, the pharmacokinetic, neuro-pharmacokinetic, and pharmacodynamic studies were performed in albino Wistar rats.

## 2. Material and Methods

### 2.1. Materials

Atomoxetine (AXT) was received as a gift sample from Panchsheel Organics Ltd. (Mumbai, India). Olive oil, coconut oil, arachis oil, castor oil, stearic acid, glycerol monostearate, palmitic acid, and carnauba wax were received from Fine chemical Industries (Mumbai, India). Tween 80, Tween 20, Span 20, carbopol 934P, and poloxamer 407 were purchased from Oxford Laboratory Thane (Mumbai, India). A dialysis bag (MWCO 12 KDa) was obtained from HiMedia (Mumbai, India). All other chemicals and solvents used were of analytical grade and procured from the laboratory.

### 2.2. Methods

#### 2.2.1. Screening of Liquid Lipids

The solubility of AXT was determined in different oils by the supersaturation method [27]. Briefly, 1 mL of different oils (viz., olive oil, coconut oil, arachis oil, and castor oil) was taken individually in an Eppendorf tube, and an excess amount of AXT was added. The developed mixture was vortexed for 5 min and allowed to shake for 72 h in an orbital shaker at 25 °C. Then, the mixture was centrifuged at 6000 rpm for 20 min. The supernatant was collected and dissolved in ethanol. The amount of drug dissolved was analyzed by a UV spectrophotometer (UV-1800, Shimadzu Co., Kyoto, Japan) at 273 nm after making a suitable dilution with ethanol.

#### 2.2.2. Screening of Solid Lipids

One gram of different solid lipids (such as GMS, stearic acid, palmitic acid, and carnauba wax) was taken into separate glass vials and melted above 5 °C (80 °C) of the melting point of each solid lipid in the water bath [28]. The excess amount of AXT was added and vortexed for 5 min. Then the mixture was shaken for 72 h in a water bath shaker to achieve supersaturation. The mixture was centrifuged at 6000 rpm for 20 min, and the supernatant was collected. Finally, the supernatant was dissolved in ethanol, and analysis was performed by a UV spectrophotometer at 273 nm after conducting a suitable dilution with ethanol to determine the amount of ATX dissolved in each lipid.

#### 2.2.3. Screening of Surfactant

The excess amount of AXT was added to a 1 mL quantity of different surfactants, viz., Tween 80, Tween 20, and Span 20, and shaken in an orbital shaker for 72 h to achieve supersaturation [28]. The mixture was centrifuged at 6000 rpm for 20 min, and the supernatant was collected. Then, the supernatant was dissolved in ethanol and analyzed by UV spectrophotometry at 273 nm for the determination of the amount of ATX dissolved in each surfactant.

#### 2.2.4. Miscibility Study

The miscibility of selected solid and liquid lipids was determined by mixing them at different ratios (9:1, 8:2, 7:3, 6:4, and 5:5) and observing them visually for any signs of precipitation and phase separation.

## 3. Experimental Design

AXT-NLC was optimized by a three-factor and three-level ((−1, 0, +1) Box–Behnken statistical design (BBD) (design expert software version 8.0.7.1) using three independent and two dependent factors. After a preliminary study, the total lipid (1.5–5.5% *w*/*v*, A), surfactant (2.5–4.5% *w*/*v*, B), and sonication time (3–9 min, C) were chosen as independent parameters, and their influence was evaluated on particle size (PS) and entrapment efficiency (EE) (independent variables) [29]. A total of 15 formulations (12 factorial points and 3 central points to evaluate the error) in different compositions were obtained from the software (Table 1). All formulations were prepared and examined for PS and EE. The values of PS and EE of all AXT-NLC formulations were put into software to analyze the different experimental models (linear, 2nd order, and quadratic). The statistical analysis was performed for different models, and the best-fit model was selected. Polynomial equations and response plots for each response were constructed for examination of the effect of ingredients on the responses.

Selection of optimized formulation:

Numerical optimization (desirability function approach) was performed to estimate the optimized composition of AXT-NLC. The optimized formulation was prepared and analyzed for PS and EE, as well as the prediction errors from the predicted and actual values of the responses. The desirability function was calculated.

### 3.1. Formulation of AXT-NLC

AXT-NLC was prepared using the melt emulsification ultrasonication method [28] as per the given composition in Table 1**.** The selected liquid (olive oil) and solid lipid (stearic acid) were mixed at 80 °C in a water bath, and then the calculated amount of AXT (50 mg) was added. The aqueous surfactant solution (Tween 80) was prepared and maintained at 80 °C. The hot aqueous surfactant solution was mixed into the melted lipid mixture at 80 °C under continuous mechanical stirring. Finally, the primary emulsion was formed and sonicated for 3–9 min using an ultra-probe sonicator (Hielscher Ultrasonicator, Teltow, Germany) at 30 s intervals in plus mode to obtain a small size. Then the AXT-NLC formulation was cooled to room temperature and stored for further analysis.

### 3.2. NLC Characterization

The dynamic light scattering (DLS) technique using a zeta sizer machine (Nano Series ZS90, Malvern Instruments, Ltd., Malvern, UK) was employed for the determination of PS, PDI, and zeta potential of NLC. Measurements were carried out by taking a 2 mL NLC of each sample diluted with deionized water. The evaluation was performed at 25 ± 2 °C and a scattering angle of 90°. The morphology was determined by scanning electron microscopy (Quanta 200 ESEM, FEI, Hillsboro, OR, USA). Briefly, the freeze-dried NLCs were placed on a metal-specific holder for the SEM analysis, and an additional iridium coating was applied before the microscopical analysis was conducted at various magnifications.

### 3.3. Entrapment Efficiency

The ultracentrifugation method was applied for the determination of the EE of AXT in AXT-NLC. A definite volume of ATX-NLC dispersion was centrifuged at 12,000 rpm at 4 °C for 20 min using a cooling centrifuge (Remi, India). The supernatant was separated, filtered using a 0.45 m syringe filter, and then subjected to analysis by UV spectrophotometry. The EE of AXT in AXT-NLC was calculated by the following equation [29].
(1)$$\%\;EE=\frac{AXT\;used-Free\;AXT}{AXT\;used}\times 100$$

### 3.4. Differential Scanning Calorimetry (DSC)

A DSC (Pyris 6 DSC, Perkin Elmer, CT, USA) instrument was used to analyze the thermogram of pure AXT and the lyophilized AXT-NLCopt formulation (AXT-NLC13). The samples (1 mg) were packed in an aluminum pan and heated over 30–300 °C (10 °C/min) in an inert nitrogen environment. A sealed, empty aluminum pan was used as a reference [23].

### 3.5. Fourier Transform Infrared Spectroscopy (FTIR) Analysis

The FTIR spectra of pure AXT and AXT-NLCopt were recorded by an FTIR instrument (Shimadzu, Tokyo, Japan). The test samples were mixed with potassium bromide (1:10) and pressed under very high pressure (2500 psig) using the mini press to form a thin pellet, which was then scanned between 400 and 4000 cm^−1^ at 25 °C and captured the spectra [23].

### 3.6. Formulation of AXT-NLC In Situ Gel

The cold method was used for the development of the in situ gel using thermosensitive polymers [30]. As depicted in Table 2, the different concentrations of thermosensitive (i.e., poloxamer 407) were taken and dissolved in cold water (4 ± 2 °C) with continuous stirring (250 rmp, 2 h) to obtain the clear solution. This solution was kept overnight in the quiescent state at 4 ± 2 °C to affect the complete dissolution of the polymer. The obtained clear poloxamer 407 solution was then mixed with a fixed concentration of carbopol 934P (a mucoadhesive agent). The optimized AXT-NLC was dispersed into a gelling solution and stored at 4 °C in the refrigerator for evaluation. Similarly, the in situ gel of ATX was prepared using poloxamer 407 for comparative analysis.

### 3.7. Evaluation of AXT-NLC In Situ Gel

#### 3.7.1. pH Analysis

The pH of the AXT-NLC in situ gel was evaluated using a pH meter (digital pH meter, Labindia, Thane, Mumbai, India). The electrode was dipped into the formulation, equilibrated for 1 min, and the pH value was recorded.

#### 3.7.2. Viscosity

The viscosity of the prepared AXT-NLC in-situ gel formulations was measured at 37 °C by a Brookfield viscometer (Brookfield RVT) using spindle LV64 at speeds of 15 rpm.

#### 3.7.3. Drug Content

The drug content for the AXT in all AXT-NLC in situ gel formulations was determined by dissolving 1 g of prepared gel in methanol and then centrifuging at 6000 rpm for 20 min, followed by filtering through a filter medium. The solution was then diluted using phosphate-buffered saline (PBS, pH 5.5) and analyzed by UV spectrophotometry at 273 nm.

#### 3.7.4. Gelling Temperature

Gelling temperature was determined by visual examination [31]. Briefly, 6 mL of the prepared AXT-NLC in situ gel was transferred into a 25 mL beaker and stirred with a magnetic bead. The beaker was placed on the thermostatic water bath, and the temperature was increased from 18 °C to 40 °C with continuous rotation at 50 rpm. The gel was then observed visually until the bead stopped rotating. The temperature at which bead rotation stopped was considered the gelation temperature.

#### 3.7.5. In Vitro Drug Release

The release of AXT from the optimized AXT-NLC in situ gel and pure AXT was determined by the dialysis bag method (MWCO 12 kDa). The pore of the dialysis bag was activated by dipping it in distilled water for 24 h before the study [7]. AXT-NLC in situ gel (equivalent to 5 mg of AXT) and pure AXT formulation (equivalent to 5 mg of AXT) were individually filled into the respected dialysis bag and dipped separately into PBS dissolution medium (250 mL, pH 5.5) at 37 °C. The dissolution medium rotated at 75 rpm on the thermostat’s magnetic stirrer. 5 ml of the released medium was taken from the beaker at definite time intervals (0, 0.5, 1, 1.5, 2, 4, 6, 8, and 12 h), and absorbance was measured by UV spectrophotometry at 273 nm. The percentage of AXT released from each formulation at each time point was calculated using Microsoft Excel. To determine the best-fitted kinetic model, the data of in vitro AXT release from optimized AXT-NLC in situ gel were fitted to various kinetic models, and the coefficient of determination (R^2^) of each model was determined and compared.

#### 3.7.6. Ex Vivo Permeation Study

The ex vivo permeation of AXT from pure AXT and optimized AXT-NLC in situ gel was performed on excised goat nasal mucosa to simulate the human nasal mucosa condition [32]. The freshly excised goat nasal mucosa was acquired from a local slaughterhouse, where all the cartilage was removed and cleaned with normal saline. The excised goat nasal mucosa was fitted between the donor and acceptor compartments of the diffusion cell (15 mL and 0.63 cm^2^), containing 15 mL of PBS (pH 5.5) in the acceptor compartment. The temperature and rotation speed were maintained at 37 ± 0.5 °C and 50 rpm during the whole study. Precaution should be taken to ensure that the membrane is not touched by the medium’s surface. Briefly, 1.5 ml of the medium was taken at fixed time intervals and replaced with the same volume of fresh medium concurrently to maintain the sink condition. The sample was filtered and analyzed by the HPLC method for the detection of the AXT concentration [33].

### 3.8. In Vivo Studies

#### 3.8.1. Animal

The in vivo study was performed on Wistar albino rats. The Institutional Animal Ethical Committee (IAEC) of Jeeva Life Sciences Hyderabad approved the research protocol. The approval number was CPCSEA/IAEC/JLS/18/07/22/047. The study was conducted as per the CPCSEA guidelines.

#### 3.8.2. Pharmacokinetic Study

This study was performed on Wistar albino rats. The rats were distributed into four groups, each containing 18 rats. The animal dose was calculated from the human dose (10 mg), and It was 0.179 mg per 200 gm of rat [17,34,35]. The optimized AXT-NLC13 in situ gel formulation (equivalent to 0.18 mg of AXT) and pure AXT (AXT-Sol in 0.9% saline phosphate buffer) were administered to respected animal groups as given below:

Group I: Received AXT-Sol intranasally (reference standard for relative bioavailability).

Group II: Received optimized AXT-NLC in situ gel (AXT-NLC13-G4) intranasally (sample).

Group III: Received AXT-NLC in situ gel (AXT-NLC13-G4) intravenously (reference standard for absolute bioavailability).

Group IV: Received AXT-NLC in situ gel (AXT-NLC13-G4) orally (reference standard for relative bioavailability).

Before the administration of each formulation, the animals were anesthetized and held in a slanted position.

#### 3.8.3. Collection of Blood Samples and AXT Extraction

The rats were anesthetized with a diethyl ether inhaler, and blood was taken at different time points (0.5, 1, 2, 3, 6, and 12 h) from the retro-orbital plexus and collected in heparinized tubes. The sample was collected from each group, containing 3 animals at each sample point. The plasma from the blood samples was separated by centrifugation at 3000 rpm for 15 min (Remi centrifuge, India) and stored at −70 °C in a deep freezer until analysis. The rats were sacrificed after collecting the blood with excess diethyl ether, followed by cervical dislocation. Then, the brain portion was separated, rinsed with saline solution, and stored at −70 °C [7,33,36]. Briefly, 0.4 mL of plasma or brain extract with 25 µL internal standard (1 µg/mL maprotiline) was mixed with 500 µL of 0.5 M PBS (pH 9.90) and diethyl ether (2 mL). The mixture was vortexed and centrifuged for 3 min at 2500 rpm to separate the organic layer. The separated organic layer was then mixed with 200 µL of 0.2 M HCl. The mixture was vortexed for 3 min, followed by centrifugation at 2500 rpm for 3 min. Finally, the organic layer was removed, and the acid residue was dried at 90 °C under a stream of nitrogen (HGC-12, Tianjin Hengao Ltd., Tianjin, China). The dried samples were reconstituted in 100 µL of mobile phase, and the concentration was analyzed by the reported HPLC method [33].

#### 3.8.4. Analysis of Extracted Plasma or Brain Samples by HPLC Technique

AXT in the blood or brain was analyzed by the HPLC method (LC 10 AD model, Shimadzu, Tokyo, Japan). The HPLC system consists of an autosampler, UV detector, and isocratic pump with a C18 column (250 × 4.6 mm, 5 µm). The mobile phase consists of acetonitrile and phosphate buffer (0.023 M, pH 6.6) at 39:61 (*v*/*v*). 0.1% TEA was used to reduce the tailing [33]. The mobile phase was filtered through a 0.45 µm membrane filter and flowed into the HPLC system at 1 mL/min. Twenty microliters of the sample was injected into the column at 25 °C and detected by a UV detector at 273 nm.

### 3.9. Pharmacokinetic and Neuropharmacokinetic Parameters Calculation

Noncompartmental analysis was used to assess the pharmacokinetic parameters of AXT from optimized AXT-NLC13 in situ gel (AXT-NLC13-G4) orally, intravenously, or intranasally and compare them with pure AXT solution intranasally. The brain and plasma concentration–time profile curves were plotted and the peak maximal concentration (Cmax) and corresponding time (Tmax) were determined. Additionally, AUC_0–12_ and AUC_0–∞_ were calculated using the linear trapezoidal method. The elimination half-life (t_1/2_), and elimination rate constant (Kel) were also calculated. Brain targeting efficiency (BTE%) was calculated by Equation (2), which allowed for the determination of drug brain partitioning [37,38].
(2)$$\%\;BTE=\frac{\left[\frac{{AUC}_{0-\infty \left(brain\right)i.n}}{{AUC}_{0-\infty \left(blood\right)i.n}}\right]}{\left[\frac{{AUC}_{0-\infty \left(brain\right)i.v}}{{AUC}_{0-\infty \left(blood\right)i.v}}\right]}\times 100$$

The brain transport percentage (% BTP) of the drug and the bioavailability (relative and absolute) of the formulation following intranasal delivery were analyzed and compared to that of pure AXT [38]. The following Equations (3)–(6) were used for the calculation of BTP and bioavailability.
(3)$$\%\;BTP=\frac{\left[{AUC}_{0-\infty \left(brain\right)i.n}\right]-F}{\left[{AUC}_{0-\infty \left(brain\right)i.n}\right]}\times 100$$
(4)$$F=\frac{{AUC}_{0-\infty \left(brain\right)i.v}}{{AUC}_{0-\infty \left(blood\right)i.v}}\times {AUC}_{0-\infty \left(blood\right)i.n}$$
(5)$$Absolute\;Bioavalability=\frac{Dose\;i.v\;X\;{AUC}_{0-\infty \left(brain\right)i.n}}{Dose\;i.n\;X\;{AUC}_{0-\infty \left(brain\right)i.v}}$$
(6)$$Relative\;Bioavalability=\frac{Dose\;oral\;X\;{AUC}_{0-\infty \left(brain\right)i.n}}{Dose\;i.n\;X\;{AUC}_{0-\infty \left(brain\right)oral}}$$

### 3.10. Pharmacodynamics Study

#### 3.10.1. Study Design for Vascular Dementia

Male albino Wistar rats were used for the study. A total of five groups were made, and each group contains six rats. The vascular dementia was induced by oral administration of L-methionine (1.7 g/kg) for 32 days [39]. On the 16th day of the study, the optimized AXT-NLC13 *in-situ* gel (AXT-NLC13-G4) was administered intranasally, and donepezil (0.1 mg/kg/day) was administered intranasally to L-methionine-treated rats. Dementia induced by L-methionine was determined using the Morris water maze (MWM) on a video tracking system (28th–32nd day) [40]. The following pattern was used for the study [41].

Group 1: Normal control (no treatment).

Group 2: Received orally L-methionine (1.7 g/kg/day) dissolved in water and given once daily by oral gavage for 4 weeks to induce dementia (disease group).

Group 3: Pure AXT solution (0.179 mg/kg/day) was administered intranasally from the 16th day of L-methionine treatment for the remaining duration (treated group-I).

Group 4: AXT-NLC13-G4 (equivalent to 0.179 mg/kg/day) was administered intranasally from the 16th day of L-methionine treatment for the remaining duration (treated group II).

Group 5: Donepezil (0.1 mg/kg/day dispersed in water) was administered intranasally from the 16th day of L-methionine treatment for the remaining duration (acting as a standard).

#### 3.10.2. Morris Water Maze (MWM) Test

Using the MWM test, spatial learning and memory abilities were evaluated. The water maze is a spherical (180 cm in diameter, 60 cm deep/high) tank that is 30 cm deep, filled with water, and set at 28.5 ± 2 °C. A platform (12.5 cm in diameter and 38 cm high) invisible to the rats (hidden platform) was set 2 cm below the water level inside the tank, with water maintained at 28.5 ± 2 °C at a height of 40 cm [42]. This platform was positioned in the left quadrant of the pool. The acquisition trial was conducted four times from the 28th to the 31st day, with a one-hour break in between each trial. As noted below, the starting position was chosen at random in a counter-balanced way. Throughout the experiment, the target quadrant (Q1) remained constant [42]. The trial ended when the rat was discovered and climbed onto the platform, and the mean escape latency time (ELT) was calculated [41].

During the acquisition trials, ELT was measured on each day, and on day 4, ELT was employed as an indication of acquisition. On the 32nd day, the retrieval trial was conducted. The platform was taken down, and the rats were put in MWM and given 120 s to explore the maze [41]. The average amount of time spent in each of the three quadrants (Q2, Q3, and Q4) was calculated, and the amount of time spent in Q1, the target quadrant, while looking for the missing platform was used as an index for recovery. The video tracking system maze master software was used to record and analyze ELT and TSTQ.

#### 3.10.3. Statistical Analysis

Design-Expert software (8.0.7.1) was used for the optimization of the formulation. GraphPad (GraphPad Prism 10.0, Boston, MA, USA) was used to compare various groups using one-way analysis of variance (ANOVA) [23]. Results were expressed as the mean ± SD. A significant difference between groups is considered at *p* < 0.05.

## 4. Results and Discussion

### 4.1. Screening of Solid and Liquid Lipids

The supersaturation method was employed for the selection of lipids that have maximum solubility. The solubility order of AXT in liquid lipids was found to be olive oil > coconut oil > arachis oil > castor oil (Figure 1A). The highest solubility of AXT was found to be in olive oil (46.23 ± 4.23 mg/g). Similarly, the solubility of AXT was determined in different solid lipids, i.e., GMS, stearic acid, palmitic acid, and carnauba wax (Figure 1B). The order of solubility of AXT in solid lipids was found to be stearic acid > GMS > palmitic acid > carnauba wax. The highest solubility of AXT was found in stearic acid (64.23 ± 4.71 mg/g). The solubility order of the AXT in surfactants was Tween 80 > Tween 20 > span 20 (Figure 1C). The maximum solubility of AXT was found to be in Tween 80 (80.04 ± 5.34 mg/g). Therefore, stearic acid, olive oil, and Tween 80 are used for the preparation of NLCs.

### 4.2. Miscibility Study

This study was carried out to evaluate the separation or crystallization of lipids after mixing solid and liquid lipids. The miscibility was checked at various ratios of solid and liquid lipids (9:1, 8:2, 7:3, 6:4, and 5:5). A 7:3 ratio did not show any signs of separation, crystallization, or oil droplets after the formation of a smear on the filter paper. So, it was used for the preparation of NLCs.

### 4.3. Optimization

AXT-NLC was optimized by design experiment software using BBD. Fifteen AXT-NLC formulations, including twelve factorial and three central points, were obtained in different compositions of total lipids (olive oil and stearic acid) (A), surfactant (B), and sonication time (C). All formulations were prepared and evaluated for dependent responses, viz., PS and EE, as depicted in Table 1. The response value of all runs was applied to the linear, 2nd order, and quadratic models and displayed as the quadratic model being the best fit for each response (*p* < 0.0001). The regression analysis of each model for both responses was analyzed, and the data are given in Table 2. An ANOVA of the best fit of both responses was calculated, and the data are depicted in Table 3. ANOVA explains the significant (*p* < 0.05) and insignificant (*p* > 0.05) effects of each variable over the responses (PS and EE). The response surface graphs were plotted for each response, which expressed the effect of multiple factors on the response (Figure 2 and Figure 3). The predicted and experimental values of the response of each run are quantitively expressed graphically in Figure 4.

### 4.4. Effect of Olive Oil, Stearic Acid, and Sonication Time over the PS of NLC

The polynomial equation of fitted quadratic model of PS is given bellow
PS (Y_1_) = +163.33 + 37.50A − 67.63B − 17.63C − 5.50AB + 11.50AC + 4.25BC + 2.21A^2^ + 1.46B^2^ − 7.54C^2^(7)

This equation showed that BC, A^2^, and B^2^ are non-significant model terms (*p* > 0.05), i.e., these do not significantly affect the PS of NLCs, and the remaining model terms exhibited a significant (*p* < 0.05) effect on PS. The quadratic model was found to be the best-fit model among all other models. The F value is 452.93 (*p* < 0.0001), and the lack of fit is insignificant (*p* > 0.05, F-value = 8.32). An insignificant lack of fit was further confirmed by values of PS after repeated experiments (Table 1, AXT-NLC13, AXT-NLC14, and AXT-NLC15). The predicted R^2^ of 0.9816 was in realistic agreement with the adjusted R^2^ (0.9966). Adequate precision was 72.59, and a low coefficient of variance (2.19%) indicated the model was significantly fitted with adequate signal. The statistical regression analysis was performed for all models, and the data are given in Table 2. The ANOVA of the optimized quadratic model was performed, which showed the significant and insignificant effects of factors over the PS (Table 3). The PS of all AXT-NLCs was in the range of 65 nm (AXT-NLC2) to 280 nm (AXT-NLC10). Equation (7) showed that the total lipids (A: olive oil and stearic acid) showed a positive effect while other factors (B: surfactant and C: sonication time) exhibited a negative effect on PC. The PS increased as the concentration of total lipid increased simultaneously. This might be due to an increase in the viscosity of the dispersion medium, thereby increasing surface tension, which tends to increase the PS of developed NLCs [43]. A comparable type of result was reported in artemether-loaded NLCs [44]. On the other hand, surfactant (Tween 80, B) concentration exhibited the opposite influence on the PS. On increasing the surfactant, the PS decreased due to decreased interfacial tension of lipid, and that helped to break down the droplets of AXT-NLCs as well as prevent the union of AXT-NLC droplets, leading to a decreased PS of AXT-NLCs [45]. Sonication time (C) increases the PS of AXT-NLCs decreased because the breaking of particles takes place. The effect of all these factors (total lipid, surfactant, and sonication time) on PS was expressed in 3D and contour plots (Figure 2).

### 4.5. Effect of Olive Oil, Stearic Acid, and Sonication Time over the EE of AXT in AXT-NLC

The polynomial equation of fitted quadratic model of EE is given bellow
EE (Y_2_) = +82.23 + 11.87A + 2.42 B − 7.29 C − 0.80AB + 0.55AC + 0.53BC − 4.82A^2^ − 1.95B^2^ − 5.59C^2^(8)

The equation displayed, AC and BC model terms, are non-significant (*p* > 0.05) (Table 3), meaning they have no significant effects on the EE of AXT in AXT-NLCs. The quadratic model was significantly (*p* < 0.05) fitted than other models. The model F-value of the quadratic model was 805.10 (*p* < 0.0001), and the lack of fit was insignificant (*p* = 0.10, F = 9.16), indicating that the model was best fitted. Further, the value of repeated experiments (Table 1, AXT-NLC13, AXT-NLC14, and AXT-NLC15) showed that there was no significant difference in the value of EE after three repetitions. It was further confirmed by the insignificant lack of fit. The predicted R^2^ of 0.9896 was found to be in realistic agreement with the adjusted R^2^ (0.9981). The quadratic model displayed high and adequate precision (94.26) and a low percentage of coefficient of variance (0.65%), revealing the model was well fitted and validated. The statistical regression analysis was performed for all models, and the data are given in Table 2. The ANOVA of the optimized quadratic model was performed, which showed the significant (*p* < 0.05) and insignificant (*p* > 0.05) effects of factors over the EE (Table 3). The EE of AXT in all AXT-NLC formulations was in the range of 90.02% (AXT-NLC4) to 52.54% (AXT-NLC12). The equation showed the total lipids (A) and surfactant (Tween 20, B) had a synergistic impact, and the sonication time (min, C) displayed an antagonistic impact on the EE of AXT in AXT-NLCs. The EE of AXT in AXT-NLCs increased with increasing total lipid concentrations due to the enhanced solubilization capacity of the drug in the lipid matrix [46]. On increasing the concentration of surfactant (Tween 80, B), the EE of AXT in AXT-NLCs increased because the partitioning between the aqueous and lipid phases decreased, allowing the drug to easily enter the lipid core matrix (NLC) [45,47]. The sonication time decreased the EE of AXT in AXT-NLCs because there may have been leaking of the drug. The effect of all these factors (total lipid, surfactant, and sonication time) on EE was expressed in 3D and contour plots (Figure 3).

### 4.6. Selection of Optimized AXT-NLC

Numerical optimization was performed to select the optimized formulation. Various compositions were obtained, but AXT-NLC13 exhibited acceptable PS and EE, so it was taken as an optimized formulation. The AXT-NLC13 has a PS of 108.1 nm (Figure 5A) and 84.12% EE. The PDI of the AXT-NLC13 was 0.217, revealing a monodispersed NLC size distribution. The zeta potential of AXT-NLC13 was found to be −42.3 (negative), indicating the particles are desegregated from each other (Figure 5B). The AXT-NLC was spherical with smooth surfaces, as observed by SEM analysis (Figure 5C). To validate the response surface methodology, the prediction error was determined. For the selected formulation, the prediction error was found to be 3.22% and 1.82% for PS and EE, respectively. The robustness of the optimized formulation was determined by the desirability function. For an optimized formulation, the desirability function was found to be 0.98. The alignment of all observed values with predicted values suggests that BBD in combination with the desirability function is a promising approach for the optimization of AXT-NLCs.

### 4.7. FTIR Analysis

FTIR was generated to study the interaction between ATX and formulation components. The FTIR spectra of AXT and AXT-NLC13 were analyzed by the KBr pellet method. The spectrum of AXT showed characteristic peaks at 3710 cm^−1^ (N–H stretching, amino group), 2997 cm^−1^ (C-H stretching of CH2 groups), 1755 cm^−1^ (C=O stretching of the carboxylic group), and 752 cm^−1^ (aromatic deformation) (Figure 6A), revealing the purity of AXT. All the characteristic peaks of AXT were observed in the formulation (AXT-NLC13) spectrum (Figure 6B), concluding the compatibility of the drug with the selected excipients at the molecular level. Similar findings were reported in a study when resveratrol-anchored NLCs were developed and FTIR was conducted to check the compatibility of the drug with the selected excipients used for the preparation of NLCs. The finding suggests that resveratrol and selected excipients are compatible at the molecular level [23]. Similar findings were also reported in other studies, such as vitamin A-NLC [48], rifabutin-NLC [49], and diacerein-NLC [50].

### 4.8. DSC Analysis

Thermograms of AXT and AXT-NLC13 were analyzed, and the result is expressed in Figure 7. The Thermogram of the AXT exhibited a sharp endothermic peak at 172.29 °C (Figure 8A). It revealed that AXT is pure and crystalline in nature. The endothermic peak of AXT in AXT-NLC13 shifted the low melting point (Figure 8B). It might be due to the mixing of API with molten excipients during the heating run, indicating a loss in crystallinity. The diacerein-loaded NLC was developed by Disha and co-workers, and their DSC results displayed the disappearance of the diacerein peak in the thermogram of the diacerein-loaded NLC, which accredited the molecular solubilization of the drug in the lipid matrix [50]. Similarly, other findings have been reported in the literature previously, viz., efavirenz-NLC [51] and rivastigmine-NLC [52].

### 4.9. Development of AXT-NLC In Situ Gel

The AXT-NLC13 incorporated in situ gel was prepared using different concentrations of thermosensitive polymer (poloxamer 407, %) and fixed concentrations of carbopol 934P (%) (Table 4).

### 4.10. Evaluation of AXT-NLC In Situ Gel

#### 4.10.1. pH, Viscosity, and Drug Content

All the AXT-NLC-incorporated in situ gels were found to be clear and transparent when observed against a black-and-white background. The pH of all AXT-NLC in situ gels was found to be 5.2 to 5.5 (Table 4), which revealed that it is compatible with nasal pH. The viscosity of all AXT-NLC in situ gel formulations was measured at 37 °C and found to be in the range of 1054 ± 13 cps (AXT-NLC13-G1) to 3209 ± 16 cps (AXT-NLC13-G5) (Table 4). The viscosity of the gel increases with increasing concentrations of the polymer [31]. The drug content of AXT in all AXT-NLC in situ gels was found to be 96.34 ± 0.84 to 98.84 ± 0.76% (Table 4).

#### 4.10.2. Gelling Temperature

Gelling temperature is a very important parameter in the case of in situ gel for intranasal delivery because, at this temperature, the sol form of the formulation is converted into gel form after administration. If the gelling temperature is greater than the nasal temperature, then the formulation may drain out of the nose, and if it is less than 22 °C, it may be a problem in the handling and transportation of the formulation [53]. The ideal gelling temperature could be 28−32 °C for intranasal delivery [54]. The gelling temperature of the all-developed AXT-NLC13 in situ gel was measured, and the data are illustrated in Table 4. It was observed as the concentration of polymer (poloxamer 407) increased and the gelling temperature decreased simultaneously. Formulations AXT-NLC13-G1 and AXT-NLC13-G2 with 13% and 14% poloxamer, respectively, did not form gel up to 42 °C. AXT-NLC13-G4 with 19% poloxamer 407 developed the gel at 28–34 °C, while the formulation AXT-NLC13-G5 (21% poloxamer) developed the gel at 26–32 °C. In the present study, the carbopol 934P concentration (0.1 *w*/*v*) was fixed. The carbopol 934P (mucoadhesive polymer) stimulates gelling temperature, which is explained in previously published reports [54,55].

Based on the studied parameters of the in situ gel formulations, the AXT-NLC13G4 was selected as an optimized formulation for intranasal delivery because it converts into gel at 28–34 °C and was used for further investigation.

#### 4.10.3. In Vitro Drug Release

The in vitro release of pure AXT and optimized in situ gel (AXT-NLC13-G4) was performed using a dialysis bag. A comparative AXT release is depicted in Figure 8A. The pure AXT release shows the fast release, i.e., 95.47 ± 2.76% in 4 h. The optimized AXT-NLC13-G4 showed a dual release pattern, i.e., initial burst release (29.68 ± 2.87%), which is good and may provide the loading dose for achieving the desired therapeutic concentration, and later sustained release (92.89 ± 3.98% in 12 h), which is beneficial to maintaining the therapeutic concentration for a longer period (Figure 9). The delayed release of the drug in the case of AXT-NLC13-G4 might be due to the incorporation of AXT-NLC13 into gels. From the results, it was concluded that the incorporation of lipid nanoparticles into gel provides an additional gel barrier for the diffusion of the drug [56]. Moreover, the sustained release pattern also demonstrated the entrapment of AXT in the solid lipid [56]. Other authors using the in situ gel system have reported similar in vitro release results. Youssef and co-workers demonstrated ciprofloxacin (CIP) release from NLC-loaded in situ gel, CIP-NLC, and CIP control gel. CIP in situ exhibited more sustained release (61.7%) as compared to CIP-NLC (82.5%) and CIP control gel (91.5%) [57]. Fahmy and co-workers developed and studied the release of flibanserin from the in situ gel and found 94% flibanserin release in 8 h as compared to flibanserin-controlled gels (98% in 4 h) [24].

AXT-NLC13-G4 exhibited a first-order release kinetic because it has a maximum R^2^ (0.9826), indicating a diffusion and dissolution release system. The release exponent value is 0.48, showing the non-Fickian release behavior of the drug from AXT-NLC13-G4.

#### 4.10.4. Ex Vivo Permeation Study

Figure 8B shows the result of the ex vivo permeation of pure AXT and optimized AXT-NLC13-G4 through excised goat nasal mucosa. The % permeation of AXT from pure AXT and AXT-NLC13-G4 was 76.26% and 35.14% in 12 h. The flux of pure AXT and AXT-NLC13-G4 was 504.37 µg/cm^2^·h and 232.41 µg/cm^2^·h, respectively. The APC of pure AXT and AXT-NLC13-G4 was found to be 1.1 × 10^−2^ cm/sec and 5.3 × 10^−3^ cm/s, respectively. The AXT-NLC13-G4 exhibited high permeation due to lipids and surfactants in the formulation as well as the consistency of the gel. The surfactant and lipids have permeation-enhancing properties that open the junction of the nasal mucosal membrane and enhance permeation. Contrarily, the slow permeation of ATX from AXT-NLC13-G4 might be due to the interaction between the polymer network and mucin. Another suggestive reason might be the higher viscosity of the gel formulation, which contributed to increased resistance and subsequently declined diffusion. A similar result was reported by Elisa and co-workers in the carbamazepine in situ formulation for enhancing brain targeting through the nasal mucosa [22]. A similar type of finding was also reported in telmisartan-loaded chitosan-coated nanoemulgel [58] and paroxetine in situ gel for brain targeting via the intranasal route [20].

#### 4.10.5. Pharmacokinetic and Neuro-Pharmacokinetic Study

The intranasal pathway is the quickest and easiest way to target the brain since it bypasses the BBB. When delivered via the intranasal route, two pathways for brain targeting are possible, i.e., extracellular and intracellular. In contrast to the intracellular system, which requires a long time (hours), the extracellular route only needs a few minutes to deliver the drug/formulation to the olfactory bulb [59].

AXT exhibited a short elimination half-life, resulting in rapid elimination after oral delivery [7]. In the present study, AXT-loaded NLCs were developed, and an optimized formulation (ATX-NLC13) was incorporated into an in situ gel. The optimized in situ formulation (AXT-NLC13-G4) was administered through i.n, iv, and oral routes, and even pure AXT was administered through i.n for comparison. For the detection of AXT in plasma and brain homogenate, the HPLC method was used [33]. The concentration of AXT in the brain and plasma was analyzed after intranasal administration of AXT-NLC13-G4 and pure AXT as well as intravenous and oral administration of AXT-NLC13-G4. The AXT concentration in plasma/brain over time was shown graphically in Figure 9. AXT-NLC13-G4 administered intranasally showed a considerably (*p* < 0.05) higher concentration of AXT in the brain than AXT-NLC13-G4 administered intravenously and orally (Figure 9A). Additionally, the AXT-NLC13-G4 formulation had a greater Cmax (173.07 ± 13.02 ng/mL) in the brain than pure AXT (99.70 ± 6.45 ng/mL) following intranasal administration. This may be due to lipid nanovesicles as well as the retention capability of in situ gel, which enhanced the ability to boost AXT nasal penetration by opening the nasal mucosal membrane [60]. The Cmax of AXT in the brain from AXT-NLC13-G4 administered intravenously and orally was 101.18 ± 9.09 ng/mL and 96.32 ± 8.00 ng/mL, respectively. Intranasal brain targeting is characterized by five main parameters, according to earlier reports published in the literature [17,38]. These parameters are: (1) the drug concentration in the brain at 0.5 h following intranasal administration of formulation and drug solution; (2) the bioavailability of formulation administered intranasally in comparison to intravenous and oral routes; (3) the drug targeting index (DTI); (4) BTE; and (5) BTP. The authenticity of the developed AXT-NLC13-G4 formulation designed for brain targeting after intranasal administration was validated in the current investigation using all five parameters. The AXT concentration in the brain 0.5 h after AXT-NLC13-G4 i.n was 75.29 ± 4.95 ng/mL higher than pure AXT i.n (31.1 ± 3.61 ng/mL). The relative bioavailability of AXT from AXT-NLC13-G4 was 1.81-fold greater than pure AXT after intranasal administration. The absolute absorption of AXT was found to be 1.59 times higher when AXT-NLC13-G4 was administered intranasally, i.e., it showed a 1.59-fold absolute bioavailability of AXT as compared to AXT-NLC13-G4 administered intravenously. Additionally, when AXT-NLC13-G4 was delivered intranasally, the relative bioavailability of AXT was 1.95 times higher than that of AXT-NLC13-G4 orally (Table 5). The DTI values for AXT-NLC13-G4 and pure AXT were found to be 2.08 and 1.40, respectively, when administered intranasally. The fraction of intranasally administered drugs that entered the brain immediately after eluding the BBB was shown by the BTP. The olfactory bulb diffuses this fraction to the brain directly. BTP was found to be 51.91% for optimized AXT-NLC13-G4 administered intranasally as compared to pure AXT (28.64%) administered via the same route. The increased BTE (207.91%) for the AXT-NLC13-G4 compared to pure AXT (140.14%) administered via the same routes (intranasally) confirmed the efficiency of the developed formulation in targeting the brain (Table 6). Comparable types of findings were reported in previously published research [23,24,25,26]. Yasmine and coworkers developed anionic and cationic NLC in situ gel formulations for intranasal delivery. It showed significant (*p* < 0.05) high bioavailability as compared to the drug solution. Additionally, anionic in situ gel displayed 1.2 times better brain targeting efficiency than cationic NLC in situ gel [25]. Usama and coworkers developed flibanserin-loaded NLCs in situ gel for intranasal delivery and found that the brain concentration of flibanserin was 6-fold higher than that of pure flibanserin after intranasal [24]. Amarjitsing and coworkers developed a resveratrol-loaded NLC in situ gel formulation for intranasal delivery. It showed a 2.5-fold higher Cmax in the brain than pure resveratrol through intranasal administration [23]. Amarjitsing and coworkers developed the NLC-based in situ gel for the intranasal delivery of donepezil. It displayed 1.3-fold higher bioavailability than the marketed formulation administered via the intranasal route [26].

#### 4.10.6. Morris Water Maze Test

The hidden platform served as a tool for tracking changes in escape latency. Even though all groups except L-methionine gradually reduced their latencies to reach the submerged platform throughout the four days of training in the Morris water maze test. On the first day, the mean escape latency between the different groups was not significantly different. On the second, third, and fourth days, the mean escape latency was significantly (*p* < 0.05) longer in the L-methionine group (group 2) as compared to other groups, viz., control, pure AXT, AXT-NLC13-G4, and donepezil (Figure 10A), indicating poor learning performance due to the intake of L-methionine. However, disrupted performance was significantly (*p* < 0.05) improved after treatment with pure AXT, AXT-NLC13-G4, and donepezil as compared to the L-methionine group (group 2). At the end of the fourth day, disrupted performance was found to be significantly (*p* < 0.05) better for AXT-NLC13-G4 as compared to pure AXT and even more than donepezil (group 5), but the difference was not significant (*p* > 0.05). The better recovery from disrupted performance might be due to the presence of the desired concentration of AXT in the brain. This might be due to the improved bioavailability of AXT after incorporation into the in situ gel. A similar finding was reported by Suzan A. Khodir, who treated L-methionine-induced vascular dementia in rats with sitagliptin [41]. Animals demonstrated a substantial difference in the probe trial, which evaluates how well the animals retained and learned the platform position during training. The time spent in the targeted quadrant was significantly (*p* < 0.001) lower for L-methionine (27.83 s). In the case of the AXT-NLC-G4 formulation, the time spent in the targeted quadrant was 46.33 s, which was significantly (*p* < 0.001) more than pure AXT (35 s) and less than donepezil (56.33 s) (Figure 10B). The better learning performance of AXT-NLC-G4 groups as compared to pure AXT groups might be due to the formulation characteristics. The developed formulation expectedly exhibited enhanced retention time at the site of administration due to its viscosity and ability to maintain the desired drug concentration at the site of action for a longer duration as compared to pure AXT [23]. Similar findings were reported by Sachdeva and its associated during the neuroprotective potential study of sesamol and solid lipid nanoparticles of sesamol in intracerebroventricular (ICV)-streptozotocin (STZ)-induced cognitive impairment. They concluded that dose-dependent restoration of cognitive deficits in ICV-STZ rats after chronic treatment with pure sesamol and sesamol-loaded solid lipid nanoparticles was possible [42]. Nahid et al. reported the effectiveness of α and β glucoside anomers of curcumin for treating Alzheimer’s disease after nose-to-brain delivery. They concluded that the escape latency and the distance to the hidden platform in the Morris water maze were significantly higher than in both the control and test groups for both glucoside anomers of curcumin [61].

## 5. Conclusions

AXT-NLC was successfully prepared and optimized. AXT-NLC exhibited nanosize and high entrapment of AXT, as well as being spherical in shape. The FTIR study showed the compatibility of the ATX with selected excipients. DSC results showed that AXT was incorporated into the NLC matrix. The optimized AXT-NLC was successfully incorporated into the in situ gel system. The optimized AXT-NLC in situ gel displayed a sustained release profile as well as higher nasal mucosal permeation than pure AXT. The AXT-NLC in situ gel displayed significantly higher (*p* < 0.05) bioavailability of AXT in the brain than pure AXT administered intranasally in rats. Further, it showed a significantly higher BTE (207.92%) than pure AXT (140.14%) after intranasal administration, revealing that a high amount of AXT reached the brain. The Morris water maze (MWM) test showed that AXT-NLC in situ gel exhibited better activity than pure AXT through intranasal delivery. These findings suggest that the NLC in situ gel is a novel carrier of AXT for transport into the brain through nose-to-brain delivery and requires further evaluation in a preclinical study.

## 6. Limitation of the Study

The main challenge is the retention of the formulation in the nasal cavity due to nasal clearance and ciliary movement. These are the first barriers to overcome when drugs are administered via the intranasal route since these issues can be overcome to a certain extent by using a nanogel drug delivery system [21].In addition, the small volume available for formulation delivery in each nostril may prevent efficient brain drug delivery [62,63].The anatomical location of the olfactory epithelium is another main limitation of this route since the dosage form must first be able to reach this site [21].Metabolic enzymes present in the olfactory mucosa must also be considered when designing a formulation for the nose-to-brain route. Consequently, intranasal formulations must be composed of biocompatible and odorless excipients to avoid rapid elimination due to mucociliary clearance and/or enzymatic degradation.Regarding nanogels, the slow responsiveness of stimuli-sensitive hydrogels, burst release during the initial swelling phase, and uncontrollable porosity are some limitations of hydrogel drug delivery [64].Many times, preclinical studies are not correlated with clinical studies due to differences in nasal anatomy between animal species and humans. To tackle these challenges, advanced mathematical models should be designed and used [21].

## Figures and Tables

**Figure 1 pharmaceutics-15-01985-f001:**
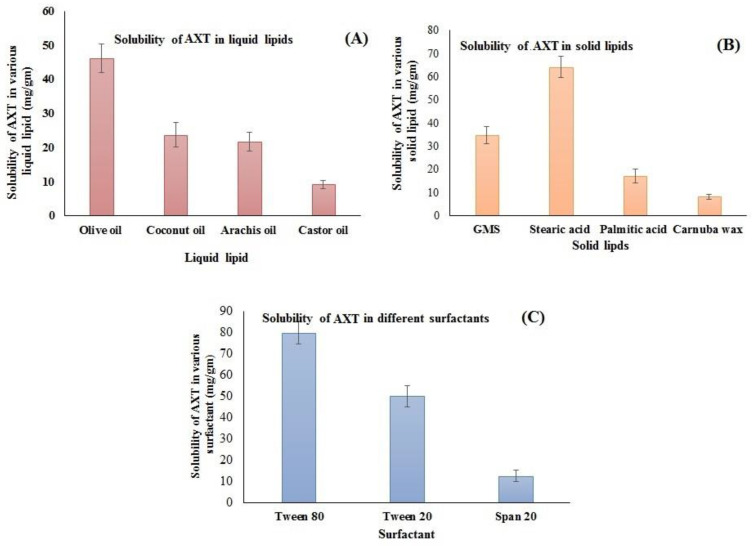
Solubility of AXT in different (**A**) liquid lipids, (**B**) solid lipids, and (**C**) surfactants. The study was performed in triplicate, and data are shown as mean ± SD.

**Figure 2 pharmaceutics-15-01985-f002:**
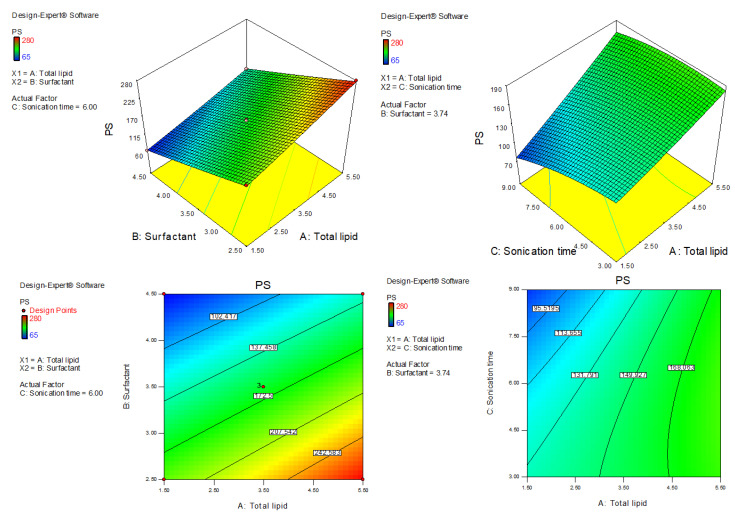
Three-dimensional and contour plots showing the effect of different independent parameters on the particle size of AXT-NLCs.

**Figure 3 pharmaceutics-15-01985-f003:**
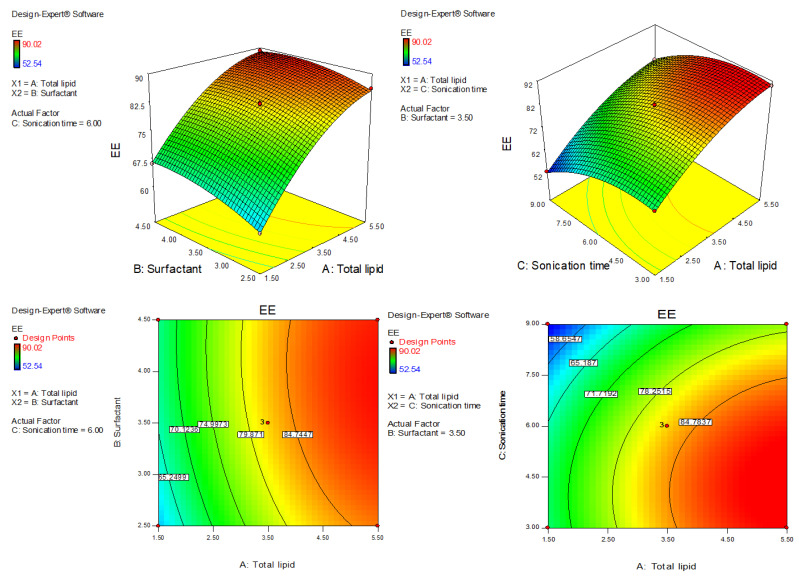
Three-dimensional and contour plots showing the effect of different independent parameters on entrapment efficiency of AXT-NLC formulations.

**Figure 4 pharmaceutics-15-01985-f004:**
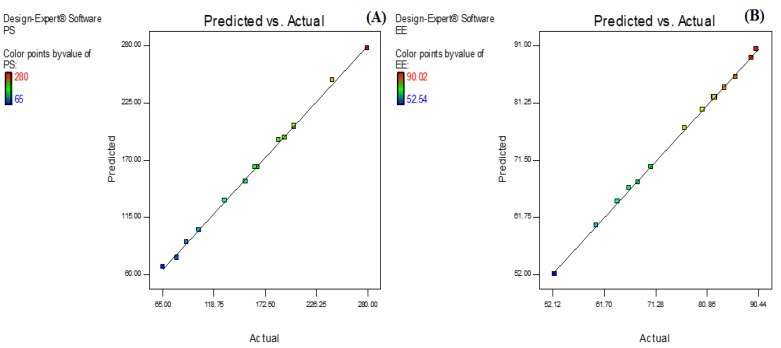
Actual and predicted value graph of (**A**) particle size and (**B**) entrapment efficiency.

**Figure 5 pharmaceutics-15-01985-f005:**
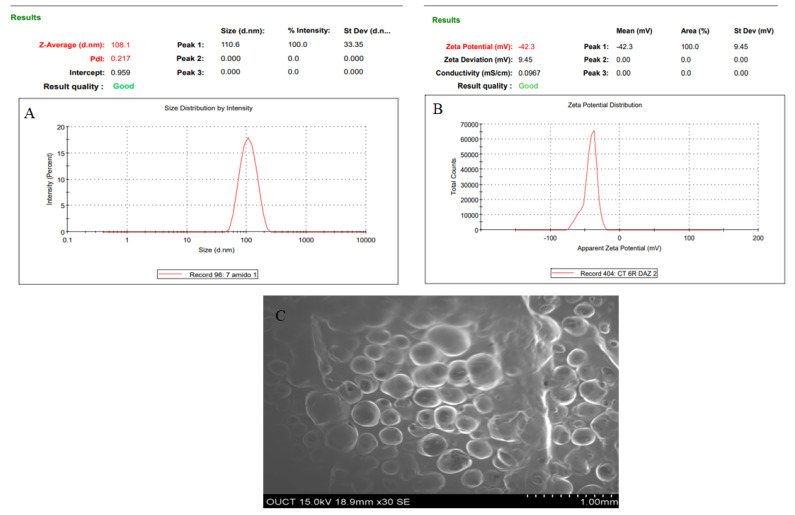
Image of optimized AXT-NLC formulation. (**A**) Particle size, (**B**) zeta potential, and (**C**) morphological evaluation by SEM.

**Figure 6 pharmaceutics-15-01985-f006:**
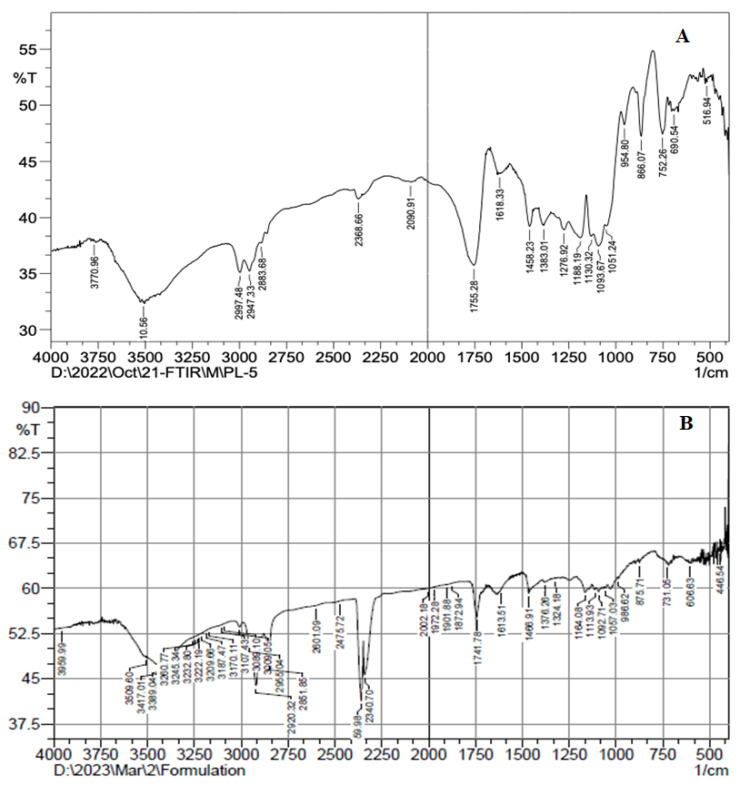
FTIR spectra of (**A**) pure AXT and (**B**) optimized formulation (AXT-NLC13).

**Figure 7 pharmaceutics-15-01985-f007:**
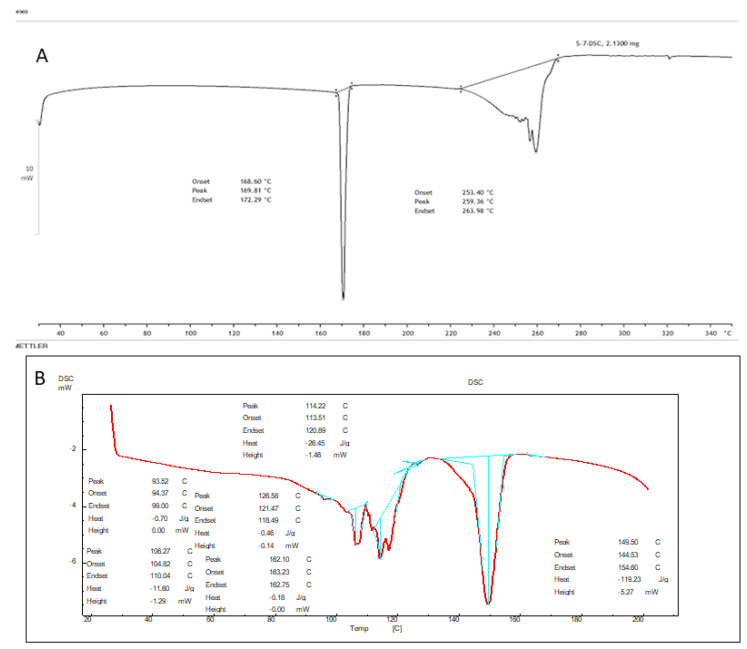
DSC image of (**A**) pure AXT and (**B**) optimized formulation (AXT-NLC 13).

**Figure 8 pharmaceutics-15-01985-f008:**
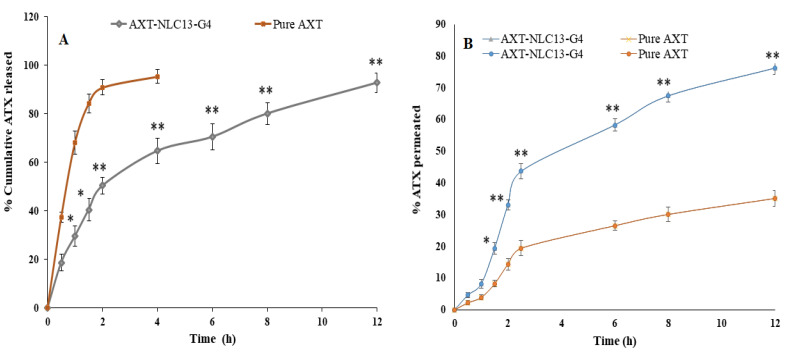
(**A**) Comparative AXT release from AXT-NLC13-Gel4 formulation and pure AXT; (**B**) ex vivo drug permeation study from AXT-NLC13-G4 formulation and pure AXT. The study was performed in triplicate, and data are shown as mean ± SD. Statistical analysis was performed between each group, and a comparison was made at *p* < 0.05. * Significant to pure AXT; ** highly significant to pure AXT.

**Figure 9 pharmaceutics-15-01985-f009:**
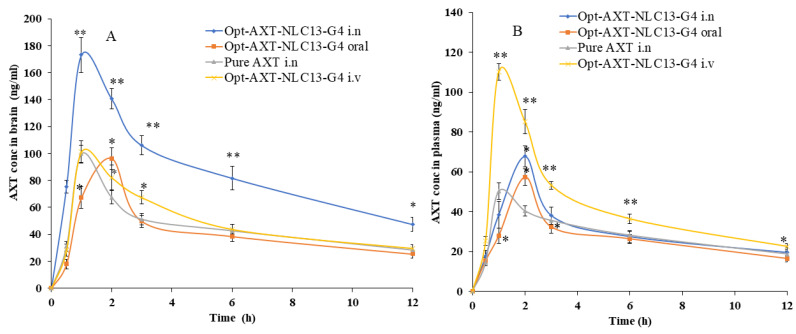
AXT concentration–time profile of the curve in (**A**) brain and (**B**) blood after intranasal, intravenous, and oral administration of the formulations (AXT-NLC13-G4 and pure AXT). The study was performed in triplicate, and data are shown as mean ± SD. Statistical analysis was performed between each group and a comparison was made at *p* < 0.05. * Significant to pure AXT; ** highly significant to pure AXT.

**Figure 10 pharmaceutics-15-01985-f010:**
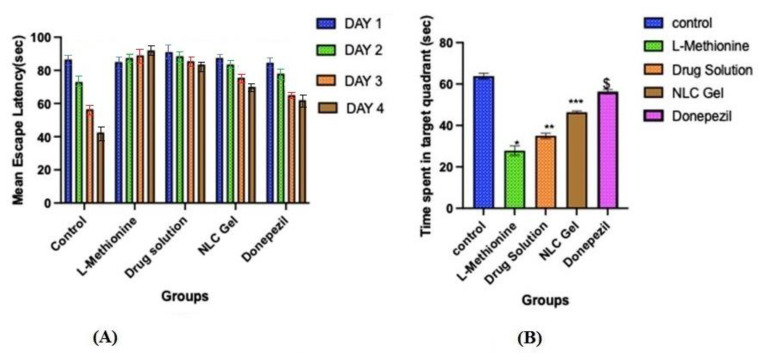
Effect of optimized AXT-NLC-G formulation on (**A**) mean escape latency and (**B**) time escape in the target quadrant. Values are expressed as mean ± SEM of n = 6 animals. * *p* < 0.001 indicates the comparison of L-methionine with the control group. ** *p* < 0.001 indicates the comparison of the drug solution with the control group. *** *p* < 0.001 indicates the comparison of NLC gel with the control group. $ *p* < 0.001 indicates the comparison of donepezil with the control group.

**Table 1 pharmaceutics-15-01985-t001:** Composition of AXT-NLC and data of PS and EE.

Formulation Code	Total Lipid (%)	Surfactant (%)	Sonication Time (min)	Particle Size (nm)	Entrapment Efficiency (%)
AXT-NLC 1	5.5	3.5	9	187.0	76.71
AXT-NLC 2	1.5	4.5	6	65.0	66.31
AXT-NLC 3	3.5	2.5	3	243.0	80.01
AXT-NLC 4	5.5	3.5	3	203.0	90.02
AXT-NLC 5	3.5	4.5	9	80.0	70.43
AXT-NLC 6	1.5	3.5	3	152.0	68.03
AXT-NLC 7	1.5	2.5	6	193.0	60.21
AXT-NLC 8	3.5	2.5	9	203.0	64.21
AXT-NLC 9	5.5	4.5	6	130.0	89.12
AXT-NLC 10	5.5	2.5	6	280.0	86.21
AXT-NLC 11	3.5	3.5	6	165.0	82.02
AXT-NLC 12	1.5	3.5	9	90.0	52.54
AXT-NLC 13	3.5	4.5	3	108.1	84.12
AXT-NLC 14	3.5	4.5	3	112.4	83.78
AXT-NLC 15	3.5	4.5	3	109.9	84.62

**Table 2 pharmaceutics-15-01985-t002:** Statistical summary of the applied models for PS and EE.

PS						
Source	Std. Dev.	R^2^	Adjusted R^2^	Predicted R^2^	PRESS	
Linear	9.70	0.9798	0.9743	0.9571	2202.77	
2FI	6.26	0.9938	0.9892	0.9688	1601.47	
Quadratic	3.54	0.9987	0.9965	0.9816	942.5	Suggested
**EE**						
Linear	4.24	0.8897	0.8596	0.8358	294.99	
2FI	4.91	0.8924	0.8117	0.7298	485.48	
Quadratic	0.49	0.9993	0.9980	0.9896	18.67	Suggested

**Table 3 pharmaceutics-15-01985-t003:** ANOVA of the best-fitted quadratic model for PS and EE.

	Particle Size (PS)				Entrapment Efficiency (EE)			
Source	Sum of Squares	df	F Value	*p*-Value Prob > F	Sum of Squares	df	F-Value	*p*-Value
Model	51,294.02	9	452.92	<0.0001	1795.81	9	805.10	<0.0001
A: Total lipid	11,250	1	894.03	<0.0001	1127.41	1	4548.98	<0.0001
B: Surfactant	36,585.13	1	2907.42	<0.0001	46.75	1	188.64	<0.0001
C: Sonication time	2485.13	1	197.49	<0.0001	424.71	1	1713.68	<0.0001
AB	121	1	9.62	0.026	2.54	1	10.26	0.0239
AC	529	1	42.04	0.001	1.18	1	4.79	0.0801
BC	72.25	1	5.74	0.062 *	1.11	1	4.49	0.0876
A^2^	18.00	1	1.43	0.285 *	85.63	1	345.51	<0.0001
B^2^	7.85	1	0.62	0.465	14.01	1	56.55	0.0007
C^2^	210.01	1	16.68	0.009	115.20	1	464.84	<0.0001
Residual	62.91	5	----	----	1.23	5	----	----
Lack of fit	58.25	3	8.32	0.1092 *	1.15	3	9.16	0.1000 *
Pure error	4.66	2	-----	------	0.08	2	-----	-----
Cor total	51,356.93	14	-----	------	1797.05	14	-----	-----

* *p* > 0.5 non-significant.

**Table 4 pharmaceutics-15-01985-t004:** Properties of AXT-NLC-incorporated gel formulations.

Batch Code	Conc of Poloxamer 407 (%)	Conc of Carbopol (%)	pH	Viscosity (cps, 37 °C)	Drug Content	Gelling Temp (°C)
AXT-NLC13G1	13	0.1	5.5	1054 ± 23	96.34 ± 0.84	No gel formation up to 42 °C
AXT-NLC13G2	15	0.1	5.2	1432 ± 26	97.45 ± 0.43	Viscosity increased but no gel formation up to 40 °C
AXT-NLC13G3	17	0.1	5.6	2109 ± 32	98.22 ± 0.24	The gel form at 36–38 °C
AXT-NLC13G4	19	0.1	5.5	2532 ± 18	98.84 ± 0.76	The gel formed at 28–34 °C
AXT-NLC13G5	21	0.1	5.3	3209 ± 26	96.85 ± 0.53	The gel formed at 26–32 °C

**Table 5 pharmaceutics-15-01985-t005:** Pharmacokinetic and bioavailability (relative and absolute) studies of optimized AXT-NLC13-G4 and pure AXT after the various routes of administration.

PP	Type of Formulation with the Route of Administration							
	AXT-NLC13-G4 i.n		AXT-NLC13-G4 i.v		AXT-NLC13-G4 Oral		AXT-Sol i.n	
	Brain	Plasma	Brain	Plasma	Brain	Plasma	Brain	Plasma
C_max_ (ng/mL)	173.07 ± 13.02 ^b,c,d^	67.72 ± 6.08	101.18 ± 9.09 ^a,c^	110.21 ± 6.64	96.32 ± 8.00 ^a,b,d^	57.39 ± 4.11	99.70 ± 6.45 ^a,c^	50.27 ± 4.09
T_max_ (h)	1.00	2.00	1.00	1.00	2.00	2.00	1.00	1
AUC_0–12_ (ng∙h/mL)	1409.58 ± 138.79 ^b,c,d^	547.86 ± 52.21	871.16 ± 82.95 ^a,c,d^	737.79 ± 72.19	720.67 ± 86.67 ^a,b,d^	469.58 ± 49.53	786.53 ± 93.86 ^a,b,c^	493.83 ± 43.18
AUC_0–∞_ (ng∙h/mL)	1520.47 ± 143.11 ^b,c,d^	627.23 ± 54.72	955.96 ± 85.58 ^a,c,d^	819.93 ± 75.03	781.04 ± 91.02 ^a,b^	514.31 ± 51.84	841.51 ± 97.21 ^a,b^	515.04 ± 45.44
AUMC_0–12_ (ng∙h^2^/mL)	11,126.77 ± 1347.33 ^b,c,d^	4979.71 ± 580.01	7617.40 ± 829.11 ^a,c,d^	6187.77 ± 779.83	5973.04 ± 924.00 ^a,b,d^	4176.95 ± 524.48	6811.74 ± 1035.80 ^a,b,c^	4227.58 ± 495.89
AUMC_0–∞_ (ng∙h^2^/mL)	14,432.88 ± 1456.36 ^b,c,d^	7403.29 ± 643.27	10,052.48 ± 894.95 ^a,c,d^	8598.35 ± 851.14	7739.61 ± 1034.50 ^a,b,d^	5461.53 ± 582.82	8341.93 ± 1119.87 ^a,b,c^	4794.36 ± 552.79
K_e_ (h^−1^)	0.17 ± 0.0015 ^b,c,b^	0.15 ± 0.0012	0.21 ± 0.0019 ^a,d^	0.19 ± 0.0027	0.73 ± 0.0025 ^a,,d^	0.21 ± 0.0034	0.26 ± 0.0021 ^a,b,c^	0.37 ± 0.0028
T1/2 (h)	4.03 ± 0.87 ^b,c,d^	4.53 ± 0.81	3.27 ± 0.70 ^a,d^	3.71 ± 0.73	3.65 ± 0.95 ^b,d^	3.27 ± 0.91	2.66 ± 0.75 ^a,b,c^	1.89 ± 0.83
RB (%) *	1.95							
RB **	1.81							
AB (%) ^#^	1.59							

All values are given as mean ± SD, n = 3, PP = pharmacokinetic parameters, RB = relative bioavailability, AB = absolute bioavailability, * relative to AXT-NLC13-G4 oral, ** relative to pure AXT i.n, ^#^ in comparison to AXT-NLC13-G4 i.v, the comparison was made at *p* < 0.05, ^a^ significantly different from the concentration of drug in the brain after i.n administration of AXT-NLC13-G4, ^b^ significantly different from the concentration of drug in the brain after i.v administration of AXT-NLC13-G4, ^c^ significantly different from the concentration of drug in the brain after oral administration of AXT-NLC13-G4, ^d^ significantly different from the concentration of drug in the brain after i.n administration of pure AXT.

**Table 6 pharmaceutics-15-01985-t006:** Neuro-pharmacokinetic study of optimized AXT-NLC13-G4 and pure AXT administered intranasally.

FC and RA	Conc of AXT at 0.5 h (ng/mL)	Brain/Blood Ratio at 0.5 h	DTI	BTE (%)	BTP (%)
AXT-NLC13-G4 i.n	75.29 ± 4.95	4.31	2.08	207.91	51.91
Pure AXT i.n	31.1 ± 3.61	2.14	1.40	140.14	28.64
AXT-NLC13-G4 i.v	28.09 ± 4.53	1.11	-	-	-
AXT-NLC13-G4 oral	17.80 ± 3.47	1.16	-	-	-

FC = formulation code, RA = route of administration, i.n = intranasal, i.v = intravenous.

## Data Availability

All data available in manuscript.

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
