# Peer review of "Development of Atomoxetine-Loaded NLC In Situ Gel for Nose-to-Brain Delivery: Optimization, In Vitro, and Preclinical Evaluation"

_pharmaceutics, 2023, doi:10.3390/pharmaceutics15071985_

Round 1

Reviewer 1 Report

The authors explored very interesting and efficient strategy for delivery of drug to the CNS. The paper is mostly well written, but it can be further improved. These are some suggestions for the authors:

1. According to the presented data, the authors did not perform repeated experiments in the central point, which is very important for assessment of the error in the lack of fit. How the adequacy of the model was assessed without calculation of the error in the repeated central point?

2. Line 422: The equation showed the total lipid (A, olive oil, and stearic acid) showed a more noticeable positive effect on PS than other factors (B and C).

This is incorrect, the effect of factor A is positive and factor B and C have negative effects.

3. Selection of optimized formulation. When use experimental design it is very useful to perform optimization in order to find optimal formulation, which may be somewhere between the formulations in the experimental design. The authors just selected the one formulation from the experimental plan which showed optimal characteristics.

4. FT-IR analysis. Comparing intensity of IR absorption bands between pure API and API in the mixture is not meaningful, due to different API content.

5. DSC analysis. Shifting of API melting peak is usually caused by mixing of API with molten excipients during heating run, rather than conversion to the amorphous form.

6. Line 562-564. It is not clear what authors consider under term slow permeation. What is used as comparison, since permeation is faster relative to pure API.

7. How the authors administered thermosensitive in-situ gel intravenously, since gelation temperature is 28-34°C. Is there possibility that gelation occurs in the bloodstream.

8. There are several mistakes in the manuscript, where the authors wrote poloxamer 405 instead of poloxamer 407.

9. Line 93. There is mistake in the last sentence. Line 127. Vertex was written instead of vortex

10. Table 2 is not placed in the right position in the text.

11. Table 4. There is mistake in the temperature labelling (-28°C)

12. Figure 9. Please check y-axis labelling

There are minor mistakes. Please see in the comments.

Author Response

Answer to Review 1 comments

The authors explored very interesting and efficient strategy for delivery of drug to the CNS. The paper is mostly well written, but it can be further improved. These are some suggestions for the authors:

Author responses against the comments/suggestions: first of all on behalf of all authors, I corresponding author, would like to thank the respected reviewer for the suggestions given and positive feedback regarding manuscripts. Our responses are:

  1. According to the presented data, the authors did not perform repeated experiments in the central point, which is very important for assessment of the error in the lack of fit. How the adequacy of the model was assessed without calculation of the error in the repeated central point?

Response: Thank you for an interesting question. Actually, we already performed repeated experiments on the central point but we did not show the data. Now as per the suggestion, we are showing it.   (Please see page 4, table 1, Page 12, & Page 13).

  1. Line 422: The equation showed the total lipid (A, olive oil, and stearic acid) showed a more noticeable positive effect on PS than other factors (B and C).

This is incorrect, the effect of factor A is positive and factors B and C have negative effects.

Response: Respected reviewer is correct. It was a typo mistake. Now we have modified the sentence as per the comment and equation 6. (Please see page 12) 

  1. Selection of optimized formulation. When use experimental design it is very useful to perform optimization in order to find optimal formulation, which may be somewhere between the formulations in the experimental design. The authors just selected one formulation from the experimental plan which showed optimal characteristics.

Response:  Thank you for the interesting question. Actually, we did numerical optimization for the selection of optimized formulation. Based on the value of PS and EE, the formulation which exhibiting acceptable value of PS & EE, its composition was similar to AXT-NLC13, so did not change the name and considered it as an optimized formulation. (Description has been given, Please see on page 13).  

  1. FT-IR analysis. Comparing intensity of IR absorption bands between pure API and API in the mixture is not meaningful, due to different API content.

Response: Thank you. FT-IR was performed for the checking the compatibility of drug with selected excipients. The description has been improved as per the question and reported literature (Please see pages 14-15)   

  1. DSC analysis. Shifting of API melting peak is usually caused by mixing of API with molten excipients during heating run, rather than conversion to the amorphous form.

Response: Thank you for the suggestion. The discussion has been updated as per the suggestion (Please see page 15)   

  1. Line 562-564. It is not clear what authors consider under term slow permeation. What is used as comparison, since permeation is faster relative to pure API.

Response: Thank you for the query. Actually, pure drug exhibited poor permeation according to the result obtained. After incorporation of ATX in NLC and gel, the permeability was increased due to the characteristic features of the formulation. A comparative permeation study was performed to evaluate the improvement in permeation after incorporation into the gel.   The explanation has been provided with suitable reference (Please see page 19)

  1. How the authors administered thermosensitive in-situ gel intravenously since gelation temperature is 28-34°C. Is there possibility that gelation occurs in the bloodstream.

Response: of course gel will develop in the bloodstream but due to regular circulation of blood, it will become diluted so it will not cause any issues.

  1. There are several mistakes in the manuscript, where the authors wrote poloxamer 405 instead of poloxamer 407.

Response: Complete manuscript has been checked and corrected for such issues (Red color highlights, please see on page 17)

  1. Line 93. There is mistake in the last sentence. Line 127. Vertex was written instead of vortex.

Response: corrected (Red color highlights, Please see on page 2) and complete manuscript also checked for further issues if any.

  1. Table 2 is not placed in the right position in the text.

Response: Now it has been placed at the right location (Please see on page 11)

  1. Table 4. There is mistake in the temperature labelling (-28°C)

Response: It was a hyphen and mistakenly placed. The correction has been made (Please see on page 18/ table 4).

  1. Figure 9. Please check y-axis labelling

Response: It was a typo mistake. Now it is okay (Please see figure 9, Page 19)

Reviewer 2 Report

Dibyalochan Mohanty et al. presented a novel carrier of AXT. Their work is original, and it may have an impact on the field. Therefore their work may be cited several times. I don't have any major objection except that the authors did not critically evaluate their work. They should provide a chapter on the limitations of their study.

Author Response

Answer to Reviewer 2 comments

Dibyalochan Mohanty et al. presented a novel carrier of AXT. Their work is original, and it may have an impact on the field. Therefore their work may be cited several times. I don't have any major objection except that the authors did not critically evaluate their work. They should provide a chapter on the limitations of their study.

Author responses against the comments/suggestions: first of all on behalf of all authors, I corresponding author, would like to thank the respected reviewer for the suggestions given and positive feedback regarding manuscripts. Our responses are:

  1. They should provide a chapter on the limitations of their study.

Response: Limitations of the study are already mentioned but based on the respected reviewer’s comments; it is further improved (Please see on Page 2, Blue color, Page 24/ heading imitation of the study, Blue color)

Reviewer 3 Report

The authors outline the formulation, characterisation, and testing of a novel nanostructured lipid drug delivery system loaded with atomoxetine as a inhalable treatment for cerebrovascular ailments such as dementia and others. This is an incredibly expansive paper loaded with details from a materials, pharmacology, and biological standpoints. The breadth and scope of the paper is impressive, however, in reviewing the article I made a number of observations. The following points must be taken into consideration by the authors when preparing a suitable revision.

1.       The writing overall could be improved. While the point being made throughout is clear, there are aspects of the writing that warrant improvement. The flow of the writing is lacking, with many parts reading like bullet points rather than structured writing. There are many typos within, and the grammar is lacking in some instances. The authors must review the entire document and address these issues.

2.       The approach to the methods section also raises questions. Why did the authors opt for a methods section, which is relatively short in content, and then a much more expansive experimental design section which also outlines several methods?    

3.       Several of the methods are vague in their descriptions. It is appreciated that this article is quite long in its scope, however, there are some methods which lack the necessary level of detail which allows one to not only repeat the experiment to mirror that of what the authors performed, but also prevents a reviewer from critically assessing the method employed and the associated rigor of such. This needs improvement.

4.       Some presentation in the figures could be improved. Several figures have writing contained within them that is incredibly difficult to make out the details of.

5.       Many figures lack error bars or any semblance of statistical analysis.

6.       The figure legends are lacking in details in some cases, and information on the n-number and the stats performed should be present in all figures with data present.  

There are several aspects of the writing that warrant attention and are highlighted in the report provided. Overall this requires extensive rewrites from a number of standpoints. 

Author Response

Answer to Reviewer 3 comments

The authors outline the formulation, characterization, and testing of a novel nanostructured lipid drug delivery system loaded with atomoxetine as an inhalable treatment for cerebrovascular ailments such as dementia and others. This is an incredibly expansive paper loaded with details from materials, pharmacology, and biological standpoints. The breadth and scope of the paper is impressive, however, in reviewing the article I made a number of observations. The following points must be taken into consideration by the authors when preparing a suitable revision.

Author responses against the comments/suggestions: first of all on behalf of all authors, I corresponding author, would like to thank the respected reviewer for the suggestions given and positive feedback regarding manuscripts. Our responses are:

  1. The writing overall could be improved. While the point being made throughout is clear, there are aspects of the writing that warrant improvement. The flow of the writing is lacking, with many parts reading like bullet points rather than structured writing. There are many typos within, and the grammar is lacking in some instances. The authors must review the entire document and address these issues.

Response: The entire document has been checked for the flow of the writing, typo errors, & grammar and corrected accordingly (All corrections are highlighted in green)

  1. The approach to the methods section also raises questions. Why did the authors opt for a methods section, which is relatively short in content, and then a much more expansive experimental design section which also outlines several methods? 

Response: The method section has been evaluated and extended as per the suggestion and need (All corrections are highlighted in green)

  1. Several of the methods are vague in their descriptions. It is appreciated that this article is quite long in its scope, however, some methods lack the necessary level of detail which allows one to not only repeat the experiment to mirror that of what the authors performed but also prevents a reviewer from critically assessing the method employed and the associated rigor of such. This needs improvement.

Response: The entire document has been checked for any vague and reading clarity has been improved by improving the language of sentences as per the suggestions (All corrections are highlighted in green)  

  1. Some presentation in the figures could be improved. Several figures have writing contained within them that is incredibly difficult to make out the details of.

Response: Figures' quality has been improved up to the possible extent.

  1. Many figures lack error bars or any semblance of statistical analysis.

Response: This manuscript contains a total of 10 figures. Figures 2, 3, 4, 5, 6, & 7 are not eligible for error bars. The rest of the figures like 1, 8, 9 & 10B contain error bars. Now we added error bars in Figure 10 A also (Please see the figure 1, 8, 9 & 10B)

  1. The figure legends are lacking in details in some cases, and information on the n-number and the stats performed should be present in all figures with data present.  

Response: Figure legends of some figures like 1, 8, and 9 have been improved. Statistical detail was also added (See Figures 1, 8, and 9) while Figure 10 already contains Statistical detail. Other figures are software-based and do not require statistical detail.

Round 2

Reviewer 3 Report

The authors have suitably addressed my comments and the manuscript is much improved. 

The manuscript is much improved. There remains some minor instances of those issues that were previously reported, but these should be easily addressed in the final proofs if the manuscript is accepted for publication.